

**Manifestations and environmental implications of microbially-induced calcium**
**carbonate precipitation (MICP) by the cyanobacterium *Dolichospermum***
***flosaquae***
Refat Abdel-Basset[1*], Elhagag Ahmed Hassan[1^] and Hans-Peter Grossart[2,3]
[1*] Botany and Microbiology Department, Faculty of Science, Assiut University, 71516
Assiut (Egypt)
[2]Dept. Experimental Limnology, Leibniz Institute for Freshwater Ecology and Inland
Fisheries, D-16775 Stechlin, Germany (hgrossart@igb-berlin.de)
[3]Dept. of Biochemistry and Biology, Potsdam University, 14469 Potsdam, Germany
^ elhagaghassan@aun.edu.eg
Corresponding author: rbasset@aun.edu.eg
**Abstract:**
The aim of this work is to explore the ability and magnitude of the temperate
cyanobacterium *Dolichospermum flosaquae* in MICP (microbially-induced calcium
carbonate precipitation). Environmentally, MICP controls the availability of calcium,
carbon and phosphorus in freshwater lakes and simultaneously controls carbon
exchange with the atmosphere. Cultures of *flosaquae* were grown in BG11 medium
containing 0, 1, 1.5, 2 and 4 mg $Ca^{2+}$ $L^{-1}$, as cardinal concentrations previously
reported in freshwater lakes, in addition to a control culture (BG11 containing 13 mg
$Ca^{2+}$ $L^{-1}$). Growth (cell number, chlorophyll a, and protein content) of *D. flosaquae* was
generally reduced by elevating calcium concentrations of the different salts used
(chloride, acetate, or citrate). *D. flosaquae* seems able to perform MICP as carbonate
alkalinity was sharply induced up to its highest level (six times that of the control) at a
citrate concentration of 4 mg $Ca^{2+}$ $L^{-1}$. Calcium carbonate was formed at a pre-
precipitation stage as the minimum pH necessary for precipitation (8.7) has been
scarcely approached under such conditions. MICP took place mostly relying on
photosynthesis and respiration, but not on urease activity, as urea was not
supplemented in the growth media. However, *D. flosaquae* exhibited strong urease
specific activity in *in vitro* assays (four times that of the control at 4 mg $Ca^{2+}$ citrate $L^{-}$
$^{1}$). Residual calcium exhibited its lowest value at 4 mg $Ca^{2+}$ citrate $L^{-1}$, coinciding with
the highest alkalinity level. Consumed calcium was increasing with chlorophyll a





content, but not with increasing cell numbers. The experiments should be repeated in
a future study, but in the presence of urea, to evaluate the full potential of calcium
carbonate precipitation of *D. flosaquae*, its dynamics and impact on biogeochemical
cycles of calcium, carbon and phosphorus in freshwater lakes.
**Key Words:**
*Dolichospermum flosaquae* – MICP - Photosynthesis – Respiration – Urease –
Alkalinity – Calcium
**Introduction:**
Microbially-induced calcium carbonate precipitation (MICP) depicts an exogenous or
endogenous microbial activity that takes place during heterotrophic growth of
numerous fungi and bacteria or during photoautotrophic growth of cyanobacteria in
their natural environments including water, soils, tufas, biofilms or geological
formations. Furthermore, bacterial, and cyanobacterial mucilaginous sheath (capsular
polysaccharides or exopolysaccharides) as well as fungal chitin act as nucleation sites
for $CaCO_3$ crystallization by binding $Ca^{2+}$ onto their carboxylic groups. MICP requires
sufficient $Ca^{2+}$, an alkaline pH and suitable microorganisms. Availability of nucleation
sites is very important for stable and continuous calcium carbonate bio-mineralization
(**Phillips** et al 2013). In particular, cyanobacteria are active prokaryotes performing
MICP (**Payandi**-Rolland 2019; **Xu** et al 2019). Several metabolic processes such as
photosynthesis, respiration, sulfate, nitrate or sulfide reduction have been recorded as
driving mechanisms for MICP (e.g. **Seifan** et al 2019). However, ureolytic activity has
been also found as a major mechanism catalyzing MICP via $CO_2$ and ammonia
production during urea hydrolysis. **Castro**-Alonso et al (2019) reported on a series of
complex reactions of urease, calcium, and carbonate during MICP surrounding the
cells. $Ca^{2+}$, beside being a component of $CaCO_3$, is inductive for urease activity
resulting in a pronounced upregulation. Furthermore, morphology of the calcite crystal
is strain-specific **(Hammes** et al 2003) and depends on the type of the calcium salt
present (**Achal** and Pan 2014).
Subsequent to precipitation of calcium carbonate, the bioavailability of both calcium
and phosphorus as well as $CO_2$ are lowered in lakes; which, in turn, arises as rate-
limiting to growth and nutrition of aquatic microbiota, e.g. via photosynthetic activity in
the case of cyanobacteria. This metabolic process is widely explored and regarded as



a promising phenomenon for use in various industrial applications. In addition to the
governmental control of acid deposition into lakes, MICP may be responsible (at least
in part) for the widespread threat of calcium decline in freshwater lakes around the
globe, as calcium co-precipitates simultaneously with carbonate (**Jeziorski** and Smol
2017). After studying thousands of water samples in tens of countries (spatially and
temporally), **Weyhenmeyer** et al (2019) concluded that the global median calcium
concentration was 4.0 mg $L^{-1}$ with 20.7% of the water samples showing $Ca^{2+}$
concentrations ≤ 1.5 mg $L^{-1}$, a threshold considered critical for the survival of many
$Ca^{2+}$ dependent organisms.
The hypothesis of this work is to explore whether *Dolichospermum flosaquae*, a major
temperate cyanobacterium, is able to perform MICP in freshwater lakes. Dependence
of MICP magnitude on $Ca^{2+}$ concentration and salt type of chloride, acetate or citrate
as well as the powering metabolic process are also concerned within this study.
Photosynthesis, respiration, total alkalinity and urease activity of *D. flosaquae* are
measurables assessed to elucidate their role in mediating MICP and to detect the
effect of the applied treatments. The results obtained are discussed on the lights of
their anticipated environmental impact and implications.

**Materials and Methods:**
Experimental Set up:
Cultures of the cyanobacterium *D. flosaquae* were incubated at different $Ca^{2+}$
concentrations (0, 1, 1.5, 2 and 4 mg $Ca^{2+} L^{-1}$) of different salts (chloride "Cl", acetate
"Ac", or citrate "Cit") supplemented into calcium free BG11 medium specific for
cyanobacteria (**Stanier** et al 1971). *D. flosaquae* was also grown in full BG11 medium
(containing 13 mg $Ca^{2+} L^{-1}$, which is considered the control culture "Con+") in addition
to a reference calcium-deprived culture (BG11 devoid of any supplemental calcium
"Con-"). Culture media were inoculated with 10 ml of 5 days old cells of *D. flosaquae*
in conical flasks capped with aluminum foil. Cultures were shaken for 4 weeks at 22±1
°C and white light intensity of 25 µmole $m^{-2} sec^{-1}$ (14h light:10h dark cycle).

Analytical Methods:



At the end of the experiment, i.e. after growth for 4 weeks, the following parameters of the variously treated *D. flosaquae* cultures were analyzed and assessed as follows:

- Cell number and chlorophyll a were simultaneously assessed using a YSI-multiparameter probe.

- Protein contents were estimated according to the method of **Bradford** (1976). Cells were extracted in boiled water, centrifuged and proteins were assessed in the supernatant The binding of protein molecules with the Coomassie Brilliant Blue dye under acidic conditions results in a color change from brown to blue, measured at a wavelength of 595 nm using a BioTek Synergy 2, multidetector microplate reader (Vermont, USA).

- Total alkalinity was assessed by titration of 50 mL algal culture media with 0.1M HCl following **Choi** et al (2017) and **Xu** et al (2019), and then calculated using the following equation:

$$CaCO_3 + 2HCl \rightarrow CaCl_2 + H_2O + CO_2$$

Based on the reaction stoichiometry between $CaCO_3$ and HCl, the mole ratio of $CaCO_3$ to HCl is 1:2; by dividing the number of moles of HCl by 2, the product is the number of moles of $CaCO_3$. The number of moles of $CaCO_3$ would be multiplied with its molecular weight to get the yielded respective $CaCO_3$ mass.

- Photosynthetic activity: The light-induced $O_2$ evolution by *D. flosaquae* in different cultures was followed by means of an oxygen sensor (PreSens MicroXTX3$O_2$ sensor, SoftwareTx3v6$O_2$, Presens, Germany) at the same growth conditions (white light intensity of about 25 µmole m$^{-2}$ sec$^{-1}$ at room temperature, i.e. 22±1ºC). Respiration ($O_2$ uptake) was also monitored using the same oxygen sensor, but in the dark.

- Assessment of residual free [$Ca^{2+}$] in the growth media: At the end of the experiment, calcium was assayed by calcium kits (ab102505, Calcium Detection Assay Kit-colorimetric, abcam) and determined at a wave length of 575 nm using a BioTek Synergy 2, multidetector microplate reader (Vermont, USA). Consumed calcium was then calculated by subtracting residual from total calcium.





- Urease enzyme (UE) activity was assayed spectrophotometrically following the
procedure of **Mobley** et al (1988) and quantified using a calibration curve of
ammonia. The in vitro assay mixture of UE contained intact cells of *D. flosaquae*,
urea (200 mM), phenol red (7 µg mL$^{-1}$) and phosphate buffer (pH 6.8). After 10 min,
the developed color, as a result of liberated ammonia from urea hydrolysis, was
determined at a wavelength of 500 nm using the same microplate reader (as
described above).

- Ammonia accumulated in the different culture media at the end of the experimental
period was assessed as mentioned above in urease-liberated ammonia.

- The pH values of the differently treated cultures were determined via a pH meter
(WTW3301, Germany).

All experiments and assessments were conducted in triplicates and the mean values
± SE (standard error) are presented in the figures.

**Results:**
Under culture conditions, growth indices (cell number, chlorophyll - and protein
contents) of *D. flosaquae* were variably affected in response to calcium concentration
as well as to its counter anion (chloride, acetate, or citrate). Growth of *D. flosaquae*
decreased as calcium concentrations of all salts were lowered, following a relative
growth enhancement (higher than the control) at a threshold value of 1.5-2.0 mg Ca$^{2+}$
L$^{-1}$ of the calcium salts citrate and acetate, respectively, while continually lowered by
calcium chloride. Calcium-deprived cultures (Co-) exhibited markedly lower growth
rates than Ca$^{2+}$ supplemented ones (Co+) in terms of cell number, chlorophyll a and
protein contents (Fig. 1).
The pH of 7.0 was set for all *D. flosaquae* cultures at the beginning of the experiment;
thereafter, it was elevated to levels ranging between pH 8.0 - 8.7, depending on
calcium treatment (Fig. 2). A certain calcium concentration of each salt induced a
higher pH than control or calcium-deprived cultures. The highest pH elevation (up to
8.7) occurred at calcium chloride and calcium citrate concentrations of 1 mg Ca$^{2+}$ L$^{-1}$



but decreased at higher concentrations. However, $Ca^{2+}$ acetate resulted in the highest
pH elevation of 8.7 at the highest added concentration (4 mg $Ca^{2+}$ $L^{-1}$). Accordingly,
the pH elevation depended on the calcium concentrations and type of $Ca^{2+}$ salt
(chloride, acetate or citrate).

Fig. (3) presents the net photosynthetic oxygen evolution ($P_N$) and dark respiratory
oxygen uptake ($R_D$) of *D. flosaquae* in dependence on the imposed calcium
treatments. $P_N$ was severely inhibited in calcium deprived cultures of *D. flosaquae*
relative to control cultures while $R_D$ was enhanced. Different calcium salts exerted
different impacts, but in most calcium treatments, net photosynthetic oxygen evolution
was higher than in the control cultures. In calcium chloride and acetate treated cultures
of *D. flosaquae*, $P_N$ and $R_D$ enhanced with increased concentrations of calcium while
in citrate treated *D. flosaquae* cultures, both $P_N$ and $R_D$ decreased.
Photosynthesis:respiration ($P_N$:$R_D$) ratios, which represent the net productivity of cells
or cultures, were severely inhibited by calcium deprivation while calcium chloride,
acetate and citrate induced inhibition or stimulation of $P_N$:$R_D$, depending on the calcium
concentration.

Total alkalinity (T alkalinity), ammonia as well as corrected carbonate alkalinity values
(C alkalinity), calculated by subtracting ammonia concentration from total alkalinity of
the differently treated *D. flosaquae* cultures, are shown in Fig. (4). Total alkalinity
exhibited its absolutely lowest amount in the control culture of *D. flosaquae*, despite it
contained the highest $Ca^{2+}$ concentration (13 mg $Ca^{2+}$ $L^{-1}$) while calcium deprivation
remarkably enhanced alkalinity up to three times that of the control cultures (5 to 15
mmol carbonate µg Chl$^{-1}$). Furthermore, alkalinity level in any of the calcium treated
cultures was markedly higher than that of control or calcium-deprived cultures, with a
maximum alkalinity level at calcium citrate concentration of 4 mg $Ca^{2+}$ $L^{-1}$. All calcium
acetate concentrations induced more or less similar alkalinity levels whereas calcium
chloride induced its highest stimulation at 2 mg $Ca^{2+}$ $L^{-1}$. As ammonia may interfere
with carbonate alkalinity, ammonia has been assessed and detected in trace amounts
not affecting total alkalinity (Fig. 4).



Residual calcium was assessed while total and consumed fractions were calculated
(per mL culture and per unit chlorophyll) and presented in Fig. (5a&b); consumed
calcium means its incorporation into or precipitation as calcium carbonate. It is
important to mention that in calcium-deprived cultures, i.e. without any external
supplementation, calcium concentration was still 2.26 mg $Ca^{2+} L^{-1}$, nevertheless. This
amount might have been released from cellular apoplastic regions as well as from
intracellular stores. Therefore, a virtual concentration of total calcium is given to
account for the externally supplemented concentration of calcium (0, 1, 1.5, 2 or 4 mg
$Ca^{2+} L^{-1}$) and the amount of calcium found at calcium-deprivation (i.e. 2.26 mg $Ca^{2+} L^{-1}$
$^{-1}$), which was assumed to be equally released by each culture. Control cultures
displayed the highest levels of all calcium fractions as they started at the highest total
virtual concentration of 15.26 mg $Ca^{2+} L^{-1}$ (i.e. 13 mg $Ca^{2+} L^{-1}$ in BG11 plus 2.26 mg
$Ca^{2+} L^{-1}$ released). On the contrary, calcium-deprived cultures exhibited the lowest
levels of all calcium fractions since no calcium had been added and thus the released
calcium was the only calcium resource.
Residual calcium (in the culture media) and consumed calcium (per unit chlorophyll
and per unit volume) increased with elevated calcium additions (Fig 5a). The lowest
amounts of residual calcium were recorded in citrate treated cultures (almost equal to
the consumed fraction and about 50% of total calcium). The concentration of 4 mg
citrate $L^{-1}$ enhanced the calcium consumption nearly up to that of the control despite
the big difference in the externally supplemented calcium concentration (4 vs. 13 mg
$Ca^{2+} L^{-1}$, respectively). In chloride and acetate, residual calcium was considerably
higher indicating less incorporation into calcium carbonate. Consumed calcium per
unit chlorophyll (C/Chl) was increasing with increasing supplemented calcium
concentration; the highest enhancement was recorded at citrate (Fig 5b).

Urease enzyme (UE) activity is presented in Fig (6); specific activity "SA" represents
the rate of enzyme activity as µmole ammonia released. $\mu g^{-1}$ protein. $min^{-1}$ while total
activity "TA" represents the rate of enzyme activity as µmole ammonia released. $mL^{-1}$
algal suspension. $min^{-1}$. Total activity is the product of specific activity per µg protein
multiplied by the amount of protein per unit volume (mL) of algal cultures. Calcium
deprivation inhibited UE activity; the magnitude of inhibition on a volume basis "TA"




was more pronounced than the enzyme specific activity "SA" because enzyme
(protein) contents were also lower. Calcium chloride induced the highest rates of UE,
total and specific activity, at 1.5 mg $Ca^{2+}$ $L^{-1}$; otherwise, it was inhibitory at lower or
higher concentrations. Calcium acetate induced the highest rates of "TA" and "SA" at
moderate concentrations of 1.5 and 2 mg $Ca^{2+}$ $L^{-}1$, both lowest and highest
concentrations of 1 and 4 mg $Ca^{2+}$ $L^{-1}$ severely inhibited the enzyme activity. Calcium
citrate induced a continuous increase in urease activity (SA) up to its "absolutely"
highest rate at 4 mg $Ca^{2+}$ $L^{-1}$ among other concentrations and salts; such highest rate
of urease activity was in accordance with the highest level of calcium consumption i.e.
calcium may be inductive to urease activity in *D. flosaquae*. The order of UE
enhancement was as follows citrate > acetate > chloride.

**Discussion:**

Our results indicate that *Dolichospermum flosaquae* is able to perform MICP
(microbially-induced calcium carbonate precipitation). Therefore, the intensive blooms
of this organism have the potential to control the overall biogeochemistry dynamics in
freshwater bodies, i.e. controlling the availability of calcium, carbon, and phosphorus
in addition to carbon emissions into the atmosphere. In this work, the capability of the
cyanobacterium *D. flosaquae* in freshwater MICP was studied at different
concentrations of three calcium salts (chloride, acetate, and citrate). It proved that
different salt types and calcium concentrations exerted different impacts on *D.*
*flosaquae* growth and metabolism. The studied concentrations (0, 1, 1.5, 2 and 4 mg
$Ca^{2+}$ $L^{-1}$) are critical depending on previous records in the literature. In this concert,
**Weyhenmeyer** et al (2019) reported that the global median calcium concentration was
4.0 mg $L^{-1}$ with 20.7% of the water samples showing $Ca^{2+}$ concentrations of ≤ 1.5 mg
$L^{-1}$, a threshold considered critical for the survival of many organisms. Growth of *D.*
*flosaquae* in terms of cell number, protein – and chlorophyll a content, was inhibited
by calcium deprivation as well as by higher concentrations of calcium. However,
concentrations of only 1.5 mg $Ca^{2+}$ $L^{-1}$ of acetate and citrate were stimulatory for *D.*
*flosaquae* growth.
*D. flosaquae* seems able to perform MICP, as inferred from alkalinity levels in the
growth media, elevated pH values, and residual vs. consumed calcium levels.



However, MICP occurred but at a pre-precipitation stage since no precipitation has
been seen by naked eyes, due to the inability of the organism to surpass the minimum
pH threshold under our experimental conditions of inactive urease due to absence of
urea (discussed later). Therefore, ammonia concentrations were found to be marginal
in the culture media. Its interference with carbonate alkalinity can be thus ruled out
indicating that the assessed alkalinity levels are substantially carbonate alkalinity.
Carbonate alkalinity exhibited the lowest levels at control cultures but increased via
calcium deprivation. However, it was induced up to its maximum level (six times that
of the control) by the highest calcium concentration of the citrate salt (4 mg $Ca^{2+} L^{-1}$).
This notion suggests that the capacity of *D. flosaquae* for carbonate formation
depends on the salt type as well as on the $Ca^{2+}$ concentration. In this respect, calcium
chloride has been recorded to be the best salt for the production of calcite by *Bacillus*
sp. among several other calcium sources used (**Achal** and Pan 2014). In this work,
however, calcium citrate apparently fits the studied organism more than chloride or
acetate.
Alkaline pH is a prerequisite for calcium carbonate formation and stability. Most calcite
precipitation occurs under alkaline conditions of pH 8.7 to 9.5 (**Ferris** et al 2003;
**Dupraz** et al 2009). When pH levels decrease, carbonates tend to dissolve rather than
precipitate (**Loewenthal** and Marais 1982). *D. flosaquae* exhibited a continuous ability
of elevating the pH of the culture medium to high pH values, which in turn, may have
caused a slow and long lag phase of growth, but favorable conditions for MICP.
However, as long as the pH of the cultures did not surpass 8.7, i.e. the lowest pH for
precipitation, carbonate has been formed but did not precipitate (see references
above: **Loewenthal** and Marais 1982; **Ferris** et al 2003; **Dupraz** et al 2009; **Gebauer**
et al 2010).
Amongst the multiple microbial metabolic activities described in the literature to
support MICP (e.g. **Anbu** et al 2016), that of *D. flosaquae* mostly relied on
photosynthesis and respiration under conditions of this work. Urease activity, the most
universal metabolic process powering MICP, is unlikely in this case, as the growth
media was not supplemented with urea. UE activity is a potentially major source of
ammonia (shifting the pH around the cell to the alkaline side) and $CO_2$ (transforming
into calcium carbonate precipitates).



UE activity is a potentially major source of ammonia (shifting the pH around the cell to
the alkaline side); meanwhile $CO_2$ and $Ca^{2+}$ transform into calcium carbonate
precipitates. Urea hydrolysis via UE activity is not complex (**Hammes** et al 2003;
**Achal** et al 2011; **Stabnikov** et al 2013). However, the high ability of *D. flosaquae* to
shift the pH to alkalinity, without urea being included in the culture medium, indicates
sources of alkalinity other than the urease-dependent ammonia production, i.e.
photosynthesis and respiration in our case. In this context, aerobic bacteria release
$CO_2$ via cell respiration, which is paralleled by an increase in pH due to ammonia
production (**Ng** et al 2012). **Hamilton** et al (2009) stated that lakes in carbonate-rich
watersheds commonly precipitate calcium carbonate as calcite; this is accelerated by
photosynthetic uptake of carbon dioxide, elevating the pH to 9–10 and reducing
concentrations of calcium and alkalinity by up to 60%. However, urea hydrolytic strains
showed higher calcite precipitation (~20–80%) in comparison with other metabolic
pathways (**Achal** et al 2009). **Okwadha** and Li (2010) reported that the amount of
$CaCO_3$ precipitation depends more on $Ca_2^+$ concentrations than urea concentrations.
The *in* vitro assay of UE (EC 3.5.1.5), per se, was also affected by calcium
concentration and salt type, i.e. it was inhibited by calcium deprivation while it exhibited
maxima at 1.5, 2 and 4 mg $Ca^{2+}$ $L^{-1}$ for chloride, acetate, and citrate, respectively.
Calcium induces UE activity; **Hammes** et al (2003) found that UE activity increased by
tenfold in the presence of 30 mM $Ca^{2+}$ relative to its absence. UE activity is related to
cell (**Ng** et al 2012), urea and calcium concentrations (**De Muynck** et al 2010), and
high pH (**Jones** et al 1982)**.** In addition, UE is only active at high pH values specific
for urea hydrolysis. It has been reported that the optimum pH for UE is 8.0, above
which the enzyme activity decreases (**Stocks**-Fischer et al 1999; **Gorospe** et al 2013).
In this work, the results indicate that UE activity of *Dolichospermum flosaquae* was
enhanced due to an increase in specific activity of the enzyme rather than to higher
cell numbers or biomass. Urease and carbonic anhydrase expression and activities
are genetically and synergistically co-regulated for MICP (**Dhami** et al 2014; **Castro**-
Alonso et al 2019).
Residual calcium was the least in citrate-treated cultures, compared with other salts
(chloride or acetate). At 4 mg $Ca^{2+}$ $L^{-1}$ of citrate, in particular, the lowest residual $Ca^{2+}$
level coincided with the highest alkalinity level, indicating its transformation to calcium
carbonate. Consumed calcium per unit chlorophyll a was increased to its highest level





also at 4 mg $Ca^{2+}$ $L^{-1}$ of citrate treated cultures. Actually, $Ca^{2+}$ is not likely utilized by metabolic processes, but accumulates outside the cells where it is readily available for $CaCO_3$ precipitation (**Silver** et al 1975). In this work, however, the unique and superior stimulating effect of 4 mg $Ca^{2+}$ $L^{-1}$ calcium citrate compared with other salts (chloride or acetate) implies intracellular intervention of calcium ions as well as the accompanying anion in the intracellular metabolism. In this respect, citrate may serve as a carbon source and internal buffer.

### Conclusions:

- *D. flosaquae*, a major representative of temperate freshwater cyanobacteria, contributes to the microbially-induced calcium carbonate precipitation (MICP) with pronounced consequences for $Ca^{2+}$ availability in freshwater lakes as well as carbon emissions to the atmosphere.
- Carbonate was formed but did not precipitate, as the organism could not increase the pH of the cultures beyond 8.7, which is considered the minimum pH value for calcite precipitation. Although it is not a precipitate, the formed calcium carbonate proves $CO_2$ and calcium sequestration.
- The mechanism(s) powering MICP seem to be photosynthesis and respiration without the participation of urease activity (as urea was not supplemented). However, UE activity elucidated a strong activity at our in vitro assays, which might maximally operate for MICP in cases of urea supplementation.
- Calcium citrate, particularly at 4 mg $Ca^{2+}$ $L^{-1}$ was the most inductive for MICP. For the emerging MICP-dependent technologies, it is therefore, recommended to apply calcium citrate because it shows outstanding enhancement of the process.
- The results can be used in modelling the environmental implications of MICP for biogeochemical cycles of calcium, carbon and phosphorus in freshwater lakes.

### Figure legends:

**Figure (1):** Growth (cell number ($10^3$ cells $mL^{-1}$), chlorophyll ($\times 10^{-3}$ µg $mL^{-1}$) and protein contents (µg $mL^{-1}$) of the cyanobacterium *Dolichospermum flosaquae* as influenced



by various calcium treatments: Control (B), 0, 1, 1.5, 2 and 4 mM of calcium chloride
(C), calcium acetate (A) and calcium citrate (Ct). Control cultures were grown in BG11
medium containing 13 mg $Ca^{2+}$ $L^{-1}$ (chloride), 0 is calcium deprived, i.e. not
supplemented with any external calcium.

**Figure (2):** pH changes of the cyanobacterium *Dolichospermum flosaquae* as
influenced by calcium treatments (as in figure 1).

**Figure (3):** Photosynthesis and respiration rates of the cyanobacterium
*Dolichospermum flosaquae* as influenced by calcium treatments (as in figure 1).

**Figure (4):** Total alkalinity (carbonate and ammonia as mmol. µg $Chl^{-1}$) of the
cyanobacterium *Dolichospermum flosaquae* as influenced by calcium treatments (as
in figure 1).

**Figure (5):** Residual (mg $L^{-1}$), total (mg $L^{-1}$) and consumed calcium (mg $L^{-1}$ or µg µg
Chl $a^{-1}$) of the cyanobacterium *Dolichospermum flosaquae* as influenced by calcium
treatments (as in figure 1).

**Figure (6):** Urease activity, T (total) and SA (Specific Activity) of the cyanobacterium
*Dolichospermum flosaquae* as influenced by calcium treatments (as in figure 1).

**Acknowledgment:**
Prof. Dr. R. Abdel-Basset sincerely thanks the Alexander von Humboldt Stiftung for
the generous financial coverage of his research stay in the lab of Prof. Dr. H. P.
Grossart, Leibiniz IGB Berlin, and thanks to the MIBI group for their help.

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

**Author contribution:**

The first author (R. A.-B.), designed the work, implemented the experiments and wrote the drafts, the second author (E.A.H.), helped in the experiments and calculated the standard errors, the third author (H.P. G.) hosted the first two authors in his lab in IGB and revised the manuscript.

**Competing interests**

There are no competing interests among authors.

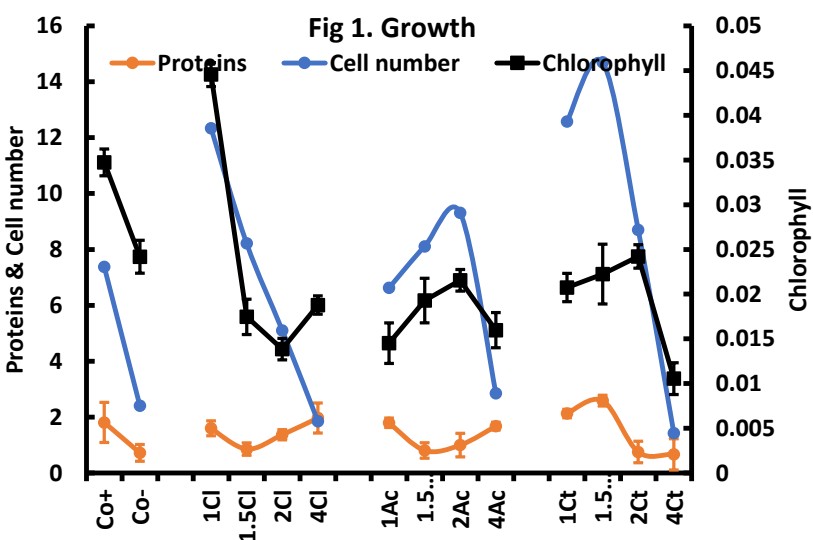

Fig 1. Growth



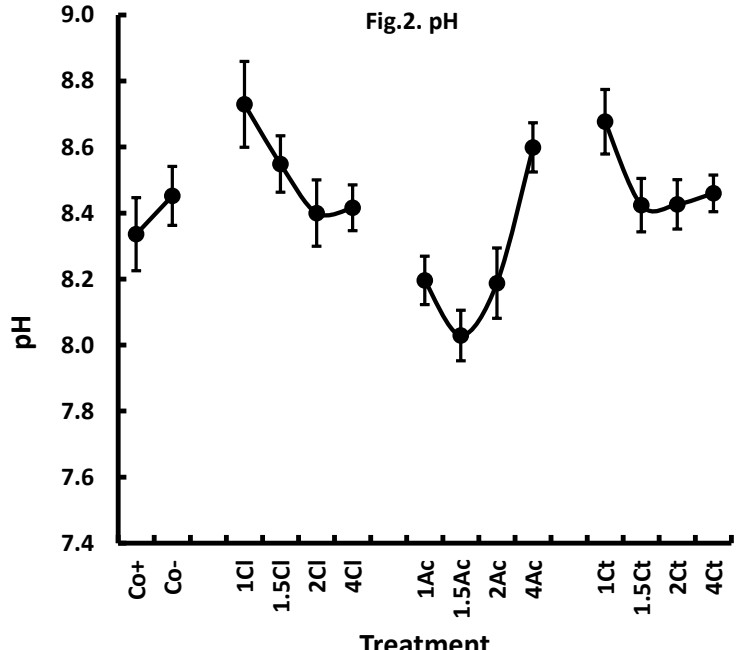





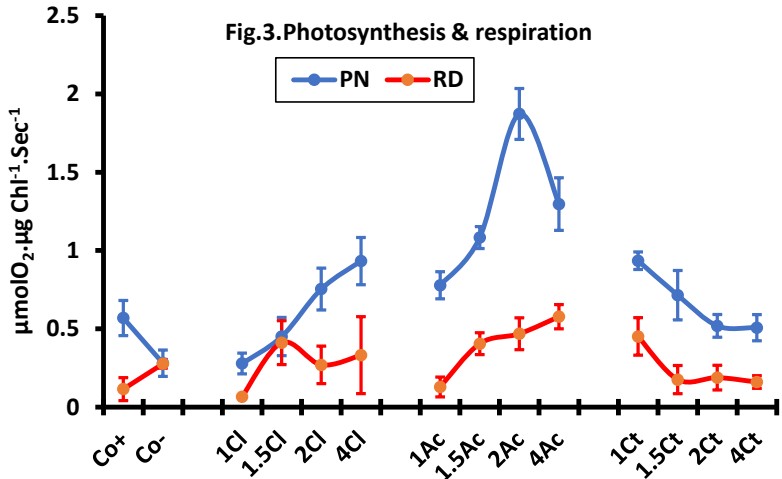

Fig.3.Photosynthesis & respiration

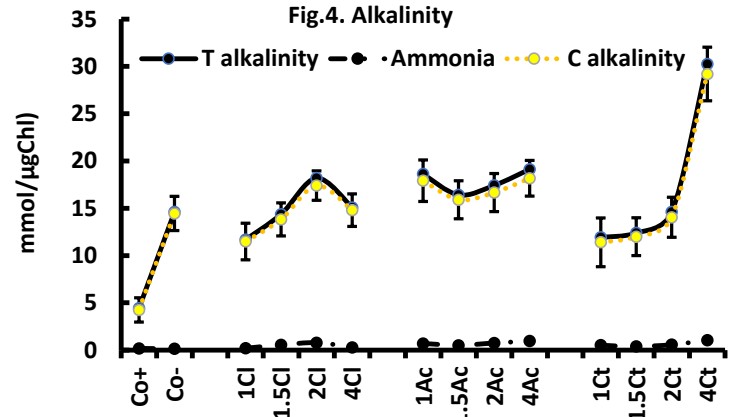

Fig.4. Alkalinity





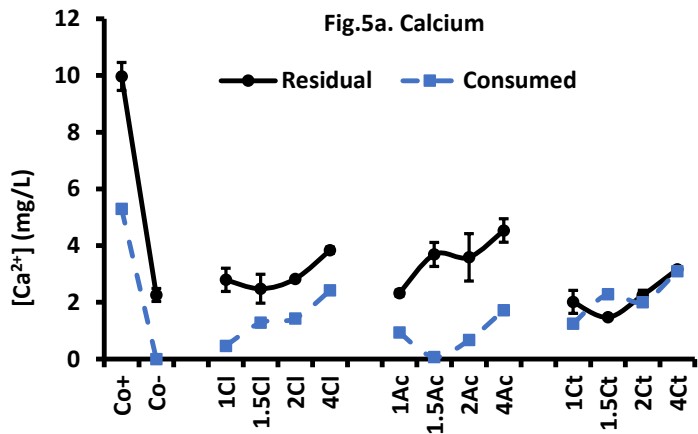

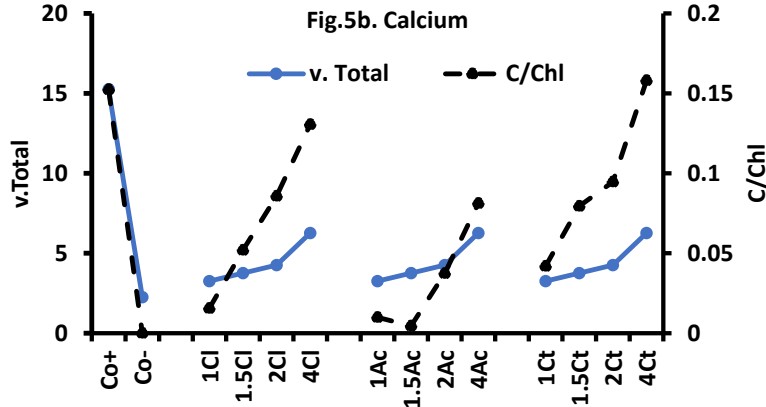



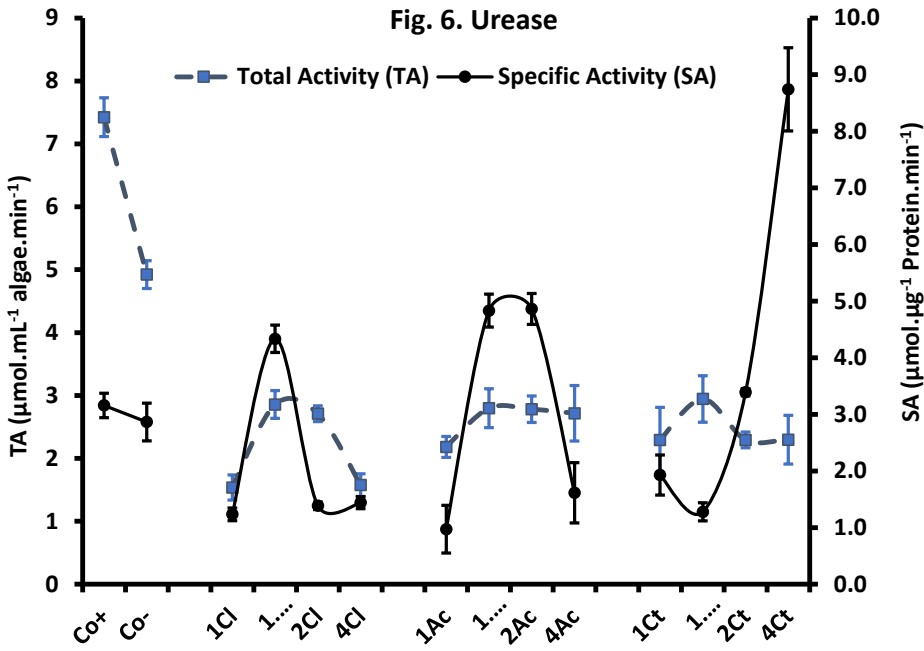