# Peer review of "Manifestations and environmental implications of microbially-induced calcium"

_Biogeosciences, 2020_

## Referee Comment (RC1) · Amal Danial (Referee) · 6 Dec 2020

In general, the overall paper is very good in quality and language. It deals with microbially-induced calcium carbonate precipitation, which has numerous environmental and commercial applications Some comments: - In the beginning of materials and methods, results and conclusion (line 85, 147 and 329) the name of the used organism must be fully written, not in its abbreviation form. - Why you started the physiological analysis after 4 weeks (it is too long period)? - It may better to mention whether you determined total or soluble proteins. - Line 102 needs full stop (.) between (supernatant)

and (The binding). - Line 130 started with "the in vitro", it can be rearranged within the sentence to avoid this beginning. - In line 349, I think you mean x103 instead of 103. - In line 352, it prefers to express chloride by "Cl" instead of "C". - In line 356, you need to rearrange the sentences to avoid beginning with small letter (pH). - In Fig 5 (line 366), you mean "residual of what?" - In line 366, Mg is repeated twice; one must be deleted. - In line 375, I think you can replace "his" with "this". - At the author contribution, it is preferred to write full names of authors. - In Fig 1 and all figures, no need to the head name of each figure (it is already written in its ligand). - In Fig 4, the extra bracket in Y axis should be deleted. - In Fig 5a, the unit in Y axis (mg/ml or mg.ml-1) should be at the same form. - In Fig 5b, letter v must be capital. - In fig 9, you may replace (ml algae) with (ml algal culture).

―――――――――――――――――

---

## Referee Comment (RC2) · Awatief Hifney (Referee) · 9 Dec 2020

The manuscript Titled: Manifestations and environmental implications of microbially-induced calcium carbonate precipitation (MICP) by the cyanobacterium Dolichospermum flosaquae(MS No.: bg-2020-378) for review is acceptable for publication in your respectable and valuable Journal (Biogeoscience ), as it deals with one of the important topics that concerns many scientists studying in the field of the environment and its changes (Biology and Biogeoscience) and its impact on the aquatic environment and the organisms that lives in it. The aquatic organisms suffer from the increase in calcium

precipitation in the lakes, which has a severe impact on the cycles of calcium, carbon and phosphorus. It is necessary to study this phenomenon using a micro-organism to know how to solve the problem described above. From my point of view, the researchers succeeded in choosing D. flosaquae, Which in return managed -to some extent- to solve the problem partly as mentioned in the manuscript result.

Best Regards and Wishes, Prof. Dr. Awatief F. Hifney Prof. of Phycology Assiut University https://www.researchgate.net/profile/Awatief_Hifney https://scholar.google.com/citations?hl=ar&user=l-nDVDEAAAAJ

————————————————————

[Figure]

**Fig. 1.**

---

## Author Comment (AC1) · 16 Dec 2020

All comments of the reviewer Amal Danial have been implemented on the text except: Prof. Dr. R. Abdel-Basset sincerely thanks the Alexander von Humboldt Stiftung for the generous financial coverage of his research stay in the lab of Prof. Dr. H. P. Grossart, Leibiniz IGB Berlin, and thanks to the MIBI group for their help. I preferred to remain "his" instead of "this" as suggested by the reviewer.

Figure (5): Residual (mg L-1), total (mg L-1) and consumed calcium (mg L-1 or $\mu$g

Chl a-1) of the cyanobacterium Dolichospermum flosaquae as influenced by calcium treatments (as in figure 1). I preferred the legend remains as such to avoid repeating the word "calcium three times in a single line to be: Residual calcium (mg L-1), total calcium (mg L-1) and consumed calcium (mg L-1 or $\mu$g Chl a-1) of......

---

## Referee Comment (RC3) · Anonymous Referee #3 · 26 Dec 2020

I have now carefully gone through the research article "Manifestations and environmental implications of microbially-induced calcium carbonate precipitation (MICP) by the cyanobacterium Dolichospermum flosaquae" authored by Refat Abdel-Basset et al. (MS No.: bg-2020-378), and so is in a position to make the following comments.

The work investigates whether the temperate cyanobacterium Dolichospermum flosaquae can induce calcium carbonate precipitation; if yes, then to what extent and under what conditions. According to the authors, microbe-induced calcium carbonate precipitation controls the availability of calcium, carbon and phosphorus in freshwater lakes and simultaneously controls carbon exchange with the atmosphere; therefore, this topic of research and the information generated have considerable significance in biogeosciences. Technically, the work has been executed by following appropriate methods and practices, and the veracity of the data presented is also quite satisfactory. However, the study has certain structural and designing-level weaknesses which need to be critically addressed before the paper can qualify as a sound geomicrobiological research work.

The term "microbe-induced", as it is used for the calcium carbonate precipitation phenomenon, indicates that the phenomenon also occurs in the absence of microbes. Several studies have highlighted microbes-independent calcite precipitation in the context of mountain springs, cave waters, hot springs and other fresh water aquatic systems. The debate on the cause and effect relationships between live microorganisms, precipitation and petrifaction is long. I think the jury is still out on whether live microbes precipitate more calcite than any other non-living micro-particulate matrix present in the aquatic system in question, and if so then should the self-inflicted burial of the causal organisms not be the limiting factor of further precipitation/mineralization within the system. In view of these issues the ecological/geomicrobiological significance of the data obtained of the present study (i.e., the scale of biomineralization rendered by the test organism Dolichospermum flosaquae) should be evaluated in relation to the scale of mineralization that is observed under abiotic conditions. Towards this end, a proper review of literature should be presented and appropriate abiotic control experiments conducted involving non-living micro-particulate matrices for calcite precipitation under various physicochemical conditions.

In relation to the choice of the test organism the present manuscript provides no rationale (the source of isolation or procurement of the cyanobacterial strain used in the present study is also not mentioned in the manuscript), on top of which we do not also get to know whether the extent of precipitation observed is high or low vis a vis precipitation levels reported previously for other cyanobacteria, fungi or bacteria, or for that the extents/rates of calcite precipitation observed over time in temperate lakes across geographies.

My other specific comments are given in the marked-up PDF file of the manuscript.

Please also note the supplement to this comment:
https://bg.copernicus.org/preprints/bg-2020-378/bg-2020-378-RC3-supplement.pdf

**Supplement:**

[revised manuscript text omitted]

---

## Author Comment (AC3) · 11 Jan 2021

Reviewer #3 I have now carefully gone through the research article "Manifestations and environ-mental implications of microbially-induced calcium carbonate precipitation (MICP) by the cyanobacterium Dolichospermum flosaquae" authored by Refat Abdel-Basset et al. (MS No.: bg-2020-378), and so is in a position to make the following comments. The work investigates whether the temperate cyanobacterium Dolichospermum flosaquae can induce calcium carbonate precipitation; if yes, then to what extent and under what conditions. According to the authors, microbe-induced calcium carbonate

precipitation controls the availability of calcium, carbon and phosphorus in freshwater lakes and simultaneously controls carbon exchange with the atmosphere; therefore, this topic of research and the information generated have considerable significance in biogeosciences. Technically, the work has been executed by following appropriate methods and practices, and the veracity of the data presented is also quite satisfactory. However, the study has certain structural and designing-level weaknesses which need to be critically addressed before the paper can qualify as a sound geomicrobiological research work. The term "microbe-induced", as it is used for the calcium carbonate precipitation phenomenon, indicates that the phenomenon also occurs in the absence of microbes. Several studies have highlighted microbes-independent calcite precipitation in the context of mountain springs, cave waters, hot springs and other fresh water aquatic systems. The debate on the cause and effect relationships between live microorganisms, precipitation and petrifaction is long. I think the jury is still out on whether live microbes precipitate more calcite than any other non-living micro-particulate matrix present in the aquatic system in question, and if so then should the self-inflicted burial of the causal organisms not be the limiting factor of further precipitation/mineralization within the system. In view of these issues the ecological/geomicrobiological significance of the data obtained of the present study (i.e., the scale of biomineralization rendered by the test organism Dolichospermum flosaquae) should be evaluated in relation to the scale of mineralization that is observed under abiotic conditions. Response: It is very hard to compare a uni-cyanobacterial culture (Dolichospermum flosaquae), grown for less than a month under controlled laboratory conditions, with a process occurring: 1) in nature, 2) by numerous collaborating consortia of microorganisms, 3) for a long-lasting time (billions of years), 4) under varied conditions of time, temperature, competition, synchronization and/or allelopathy. Under natural conditions, the precipitation of carbonates occurs very slowly over long geological times but in order to produce large amounts of carbonates shortly there is need to look for microbes with the ability to create conditions for precipitation of carbonates in shorter times (Dhami et al 2013). Stocks-Fischer et al (1999) reported that at pH 9.0, only 35 and 54 %

precipitated chemically in water and medium, respectively but 98 % of the initial Ca2+ concentrations were precipitated microbially. Berry et al (2002) reported that though the oceans are supersaturated with Ca2+ and CO32-, spontaneous precipitation of CaCO3 in the absence of calcifying (micro)- organisms is rare owing to various kinetic barriers. Thus, the process in nature is inefficient and the existence of a microorganism or part of it (cell walls, spores or mucilage) is indispensable for efficient calcification. It has also been reported that the largest share of global calcification takes place via biotic processes in the oceans (Olajire 2013). Reviewer #3: Towards this end, a proper review of literature should be presented, and appropriate abiotic control experiments conducted involving non-living micro-particulate matrices for calcite precipitation under various physicochemical conditions. Response: Done. Reviewer #3: In relation to the choice of the test organism the present manuscript provides no rationale (the source of isolation or procurement of the cyanobacterial strain used in the present study is also not mentioned in the manuscript), Response: now mentioned in the text as a common cyanobacterium isolated from the temperate freshwater lake Stechlinsee, Germany. Reviewer #3: On top of which we do not also get to know whether the extent of precipitation observed is high or low vis a vis precipitation levels reported previously for other cyanobacteria, fungi or bacteria, or for that the extents/rates of calcite precipitation observed over time in temperate lakes across geographies. Response: Microbially mediated calcification can be traced back for at least 2.6 billion years (Altermann et al 2006). They proposed that the interplay of cyanobacteria and heterotrophic bacteria has been the major contributor to the carbonate factory for the last 3 billion years of Earth history. For the great majority of calcium carbonate precipitations, qualitative and descriptive assessments are dominant while quantitative assessments are scarce. MICP quantities of precipitated calcium after six treatments to Bacillus sp. were 0.15 and 0.60g of Ca per cm2 of treated sand surface for the cases of bulk or surface MICP, respectively (Chu et al 2012). Also, a putative calcium carbonate mineral mass of 2.5 mg/OD 660 has been reported in Bacillus sp. JH7 (Kim et al 2017). Reviewer #3: My other specific comments are given in the marked-up PDF file of the manuscript. Please

also note the supplement to this comment:https://bg.copernicus.org/preprints/bg-2020-378/bg-2020-378-RC3-supplement.pdf Interactive comment on Biogeosciences Discuss., https://doi.org/10.5194/bg-2020-378, 2020.C3 Reviewer #3: The actual terminology should come first and then the abbreviation in parenthesis. Response: Done Reviewer #3: Why say "seems"? Was it not confirmatory? Response: changed to "exhibited its ability"

Reviewer #3: I think it should be "increased". Response: changed to "increased". Reviewer #3: What is the point in saying that MICP dd not take place on urease activity when urea was not provided in the medium in the first place? Response: Because urease activity is a considerable metabolic activity among various metabolic activities empowering MICP; the sentence has been reformulated in the text. Reviewer #3: It is not proper to write "consumed calcium". Consumption is related to either assimilation or dissimilatory energy harnessing. In this case it is simply "precipitation". Response: OK, "precipitated" substituted "consumed" in the text

Reviewer #3: was increasing Response: Changed to "consumed" in the text

Reviewer #3: Why do the authors not perform this simple experiment within this study itself? Why do they want to keep the urea-based growth and precipitation test up for future? Response: It is not to conduct this assay only. We meant to repeat the whole work i.e. growth, photosynthesis, respiration, assays, etc., in the presence of urea, in addition to some modifications and improvements based on the obtained findings.

Reviewer #3: This entire portion, though full of existing information, has got no reference cited. All the known facts and notions mentioned here need to be supported by appropriate reference(s).

Response: This is a collective sentence of mine; references are cited at their respective places throughout the detailed description.

Reviewer #3: Make sure whether you mean "MICP, simply, occurs under these

metabolic conditions" or these pathways are biochemically linked with MICP at the level of metabolites, intermediates, enzymes, etc. If you mean the latter, then elaborate explanations must be given for each biochemical connection, quite in the same way as you have given for ureolysis. Also cite original papers for each metabolism, and not leave it on a review for the reader to cross-refer from. Response: Reformulated in the new version to reveal the idea of the reviewer, but "elaborate explanation" for each biochemical connection renders the introduction too lengthy.

Reviewer #3: How does the bioavailability of calcium, phosphorus as well as CO2 get lowered in lakes together? Please describe the (bio?) geochemistry of this with proper citations. Response: reformulated in the new version to the following: Subsequent to coprecipitation of calcium and carbon(ate), chemically and/or microbially to form calcium carbonate, the bioavailability of both calcium and carbon becomes limited. Calcium and phosphate also coprecipitate and thus get lowered at these conditions. Reviewer #3: Meaning of this is not clear. Please clarify/elaborate.

Response: reformulated in the new version

Reviewer #3: What type of water bodies were included in this study? Please mention a few examples to implicate the range of environmental diversity covered.

Response: reformulated to the following: After studying 440,599 water samples from 43,184 inland water sites in 57 American and European countries, Weyhenmeyer et al (2019) concluded that the global median calcium concentration was 4.0 mg L$-1$ with 20.7% of the water samples showing Ca2+ concentrations $\leq$ 1.5 mg L$-1$, a threshold considered critical for the survival of many Ca2+ dependent organisms e.g. Daphnia (Jeziorski et al 2014). Reviewer #3: Please give an assortment of examples. Response: given, Daphnia Reviewer #3: Please give strain name and source of isolation or procurement. Response: It is a local isolate of Stechlinsee, Germany

Reviewer #3: Is this the first report of calcite precipitation by D. flosaquae ? Please clarify. Response: Yes, it is the first time: Our results indicate, for the first time, that

Dolichospermum flosaquae is able to perform MICP.

Reviewer #3: As already mentioned, we need a clear narrative on how these elements are biogeochemically interlinked in fresh water aquatic systems. Response: already done Reviewer #3: critical with respect to what? Response: critical for the survival of many Ca2+ dependent organisms (Jeziorski et al 2014). Reviewer #3: What does the previous records say? Clarity is needed on what the authors want to mean here. Response: Clarified in the text Reviewer #3: Please clarify what criticality you conclude regarding the precipitation level rendered by D. flosaquae in the context of these values. Do you mean that D. flosaquae can turn the system fully devoid of calcium in certain water bodies? Response: The results indicate decreased levels of residual calcium but not to zero. However, it is out of context in this respect. Reviewer #3: per liter of acetate and citrate? That's meaningless. Do you mean, calcium per liter in the form of acetate and citrate? Response: Yes, we mean calcium acetate and citrate per liter.

Reviewer #3: For this, please cite data from the present study. Response: Done

Reviewer #3: Meaning of this is not clear. Please clarify/elaborate. Response: reformulated as follows: Anthropogenic activities, namely acid deposition, is detrimental to calcium decline. Since some time ago, governments determined to prevent acid deposition into lakes; acid deposition solubilizes calcium i.e. no acid deposition means no calcium dissolution (Korosi et al 2012). Another explanation is that the acid deposition before such determinations may have led to depletion of calcium in soil catchments leaving no more of the element to dissolve.

---

## Author Comment (AC4) · 11 Jan 2021

Reviewer #2 BGD Interactive comment Printer-friendly version Discussion paper Biogeosciences Discuss.,https://doi.org/10.5194/bg-2020-378-RC2, 2020© Author(s) 2020. This work is distributed under the Creative Commons Attribution 4.0 License.Interactive comment on "Manifestations and environmental implications of microbially-induced calcium carbonate precipitation (MICP) by the cyanobacterium Dolichospermum flosaquae" by Refat Abdel-Basset et al.Awatief Hifney (Referee) hifney@aun.edu.eg

The manuscript Titled: Manifestations and environmental implications of microbially-induced calcium carbonate precipitation (MICP) by the cyanobacterium Dolichospermum flosaquae (MS No.: bg-2020-378) for review is acceptable for publication in your respectable and valuable Journal (Biogeoscience ), as it deals with one of the important topics that concerns many scientists studying in the field of the environment and its changes (Biology and Biogeoscience) and its impact on the aquatic environment and the organisms that lives in it. The aquatic organisms suffer from the increase in precipitation in the lakes, which has a severe impact on the cycles of calcium, carbon and phosphorus. It is necessary to study this phenomenon using a microorganism to know how to solve the problem described above. From my point of view, the re-searchers succeeded in choosing D. flosaquae, which in return managed -to some extent- to solve the problem partly as mentioned in the manuscript result. Response: Done

---

## Author Comment (AC5) · 11 Jan 2021

Reviewer #1: In the beginning of materials and methods, results and conclusion (line 85, 147 and 329) the name of the used organism must be fully written, not in its abbreviation form. - Why you started the physiological analysis after 4 weeks (it is too long period)? - It may better to mention whether you determined total or soluble proteins. - Line 102 needs full stop (.) between (supernatant) and (The binding). - Line 130 started with "the in vitro", it can be rearranged within the sentence to avoid this beginning. - In line 349, I think you mean x103 instead of 103. -In line 352, it prefers

to express chloride by "Cl" instead of "C". - In line 356, you need to rearrange the sentences to avoid beginning with small letter (pH). - In Fig 5 (line 366),you mean "residual of what?" - In line 366, Mg is repeated twice; one must be deleted.- In line 375, I think you can replace "his" with "this". - At the author contribution, it is preferred to write full names of authors. - In Fig 1 and all figures, no need to the head name of each figure (it is already written in its ligand). - In Fig 4, the extra bracket in Y axis should be deleted. - In Fig 5a, the unit in Y axis (mg/ml or mg.ml-1) should beat the same form. - In Fig 5b, letter v must be capital. - In fig 9, you may replace (ml algae) with (ml algal culture). Response: All comments of the reviewer Amal Danial have been implemented on the text, except: I preferred to remain "his" instead of "this" as suggested by the reviewer. Figure (5): Residual (mg L-1), total (mg L-1) and consumed calcium (mg. L-1 or mg. $\mu$g Chl-1) of the cyanobacterium Dolichospermum flosaquae as influenced by calcium treatments (figure 1). I prefer the legend remains as it is to avoid repeating the word "calcium three times in a single line to be: Residual calcium (mg L-1), total calcium (mg L-1) and consumed calcium (mg L-1 or $\mu$g Chl a-1) of.....